# DriPP: Driven Point Processes to Model Stimuli Induced Patterns in M/EEG Signals

**Cédric Allain, Alexandre Gramfort & Thomas Moreau**
Université Paris-Saclay, Inria, CEA, Palaiseau, 91120, France
`{cedric.allain, alexandre.gramfort, thomas.moreau}@inria.fr`

## Abstract

The quantitative analysis of non-invasive electrophysiology signals from electroencephalography (EEG) and magnetoencephalography (MEG) boils down to the identification of temporal patterns such as evoked responses, transient bursts of neural oscillations but also blinks or heartbeats for data cleaning. Several works have shown that these patterns can be extracted efficiently in an unsupervised way, *e.g.,* using Convolutional Dictionary Learning. This leads to an event-based description of the data. Given these events, a natural question is to estimate how their occurrences are modulated by certain cognitive tasks and experimental manipulations. To address it, we propose a point process approach. While point processes have been used in neuroscience in the past, in particular for single cell recordings (spike trains), techniques such as Convolutional Dictionary Learning make them amenable to human studies based on EEG/MEG signals. We develop a novel statistical point process model – called driven temporal point processes (DriPP) – where the intensity function of the point process model is linked to a set of point processes corresponding to stimulation events. We derive a fast and principled expectation-maximization (EM) algorithm to estimate the parameters of this model. Simulations reveal that model parameters can be identified from long enough signals. Results on standard MEG datasets demonstrate that our methodology reveals event-related neural responses – both evoked and induced – and isolates non-task-specific temporal patterns.

## 1 Introduction

Statistical analysis of human neural recordings is at the core of modern neuroscience research. Thanks to non-invasive recording technologies such as electroencephalography (EEG) and magnetoencephalography (MEG), or invasive techniques such as electrocorticography (ECoG) and stereotactic EEG (sEEG), the ambition is to obtain a detailed quantitative description of neural signals at the millisecond timescale when human subjects perform different cognitive tasks (Baillet, 2017). During neuroscience experiments, human subjects are exposed to several external stimuli, and we are interested in knowing how these stimuli influence neural activity.

After pre-processing steps, such as filtering or Independent Component Analysis (ICA; Winkler et al. 2015) to remove artifacts, common techniques rely on epoch averaging – to highlight evoked responses – or time-frequency analysis to quantify power changes in certain frequency bands (Cohen, 2014) – for induced responses. While these approaches have led to numerous neuroscience findings, it has also been criticized. Indeed, averaging tends to blur out the responses due to small jitters in time-locked responses, and the Fourier analysis of different frequency bands tends to neglect the harmonic structure of the signal, leading to the so-called "Fourier fallacy" (Jasper, 1948; Jones, 2016). In so doing, one may conclude to a spurious correlation between components that have actually the same origin. Moreover, artifact removal using ICA requires a tedious step of selecting the correct components.

Driven by these drawbacks, a recent trend of work aims to go beyond these classical tools by isolating prototypical waveforms related to the stimuli in the signal (Cole & Voytek, 2017; Dupré la Tour et al., 2018; Donoghue et al., 2020). The core idea consists in decomposing neural signals as combinations of time-invariant patterns, which typically correspond to transient bursts of neural activity (Sherman

et al., 2016), or artifacts such as eye blinks or heartbeats. In machine learning, various unsupervised algorithms have been historically proposed to efficiently identify patterns and their locations from multivariate temporal signals or images (Lewicki & Sejnowski, 1999; Jost et al., 2006; Heide et al., 2015; Bristow et al., 2013; Wohlberg, 2016b), with applications such as audio classification (Grosse et al., 2007) or image inpainting (Wohlberg, 2016a). For neural signals in particular, several methods have been proposed to tackle this task, such as the sliding window matching (SWM; Gips et al. 2017), the learning of recurrent waveforms (Brockmeier & Príncipe, 2016), adaptive waveform learning (AWL; Hitziger et al. 2017) or convolutional dictionary learning (CDL; Jas et al. 2017; Dupré la Tour et al. 2018). Equipped with such algorithms, the multivariate neural signals are then represented by a set of spatio-temporal patterns, called *atoms*, with their respective onsets, called *activations*. Out of all these methods, CDL has emerged as a convenient and efficient tool to extract patterns, in particular due to its ability to easily include physical priors for the patterns to recover. For example, for M/EEG data, Dupré la Tour et al. (2018) have proposed a CDL method which extracts atoms that appertain to electrical dipoles in the brain by imposing a rank-1 structure. While these methods output characteristic patterns and an event-based representation of the temporal dynamics, it is often tedious and requires a certain domain knowledge to quantify how stimuli affect the atoms' activations. Knowing such effects allows determining whether an atom is triggered by a specific type of stimulus, and if so, to quantify by how much, and with what latency.

As activations are random signals that consist of discrete events, a natural statistical framework is the one of temporal point processes (PP). PP have received a surge of interest in machine learning (Bompaire, 2019; Shchur et al., 2020; Mei et al., 2020) with diverse applications in fields such as healthcare (Lasko, 2014; Lian et al., 2015) or modelling of communities on social networks (Long et al., 2015). In neuroscience, PP have also been studied in the past, in particular to model single cell recordings and neural spike trains (Truccolo et al., 2005; Okatan et al., 2005; Kim et al., 2011; Rad & Paninski, 2011), sometimes coupled with spatial statistics (Pillow et al., 2008) or network models (Galves & Löcherbach, 2015). However, existing models do not directly address our question, namely, the characterization of the influence of a deterministic PP – the stimuli onsets – on a stochastic one – the neural activations derived from M/EEG recordings.

In this paper, we propose a novel method – called driven point process (DriPP) – to model the activation probability for CDL. This method is inspired from Hawkes processes (HP; Hawkes 1971), and models the intensity function of a stochastic process conditioned on the realization of a set of PP, called *drivers*, parametrized using truncated Gaussian kernels to better model latency effects in neural responses. The resulting process can capture the surge of activations associated to external events, thus providing a direct statistical characterization of how much a stimulus impacts the neural response, as well as the mean and standard deviation of the response's latency. We derive an efficient expectation-maximization (EM) based inference algorithm and show on synthetic data that it reliably estimates the model parameters, even in the context of MEG/EEG experiments with tens to hundreds of events at most. Finally, the evaluation of DriPP on the output of CDL for standard MEG datasets shows that it reveals neural responses linked to stimuli that can be mapped precisely both in time and in brain space. Our methodology offers a unified approach to decide if some waveforms extracted with CDL are unrelated to a cognitive task, such as artifacts or spontaneous brain activity, or if they are provoked by a stimulus – no matter if they are 'evoked' or 'induced' as more commonly described in the neuroscience literature (Tallon-Baudry et al., 1996). While these different effects are commonly extracted using different analysis pipelines, DriPP simply reveals them as stimuli-induced neural responses using a single unified method, that does not require any manual tuning or selection.

## 2   DRIVEN TEMPORAL POINT PROCESS (DRIPP)

A temporal point process (PP) is a stochastic process whose realization consists of discrete events $\{t_i\}$ occurring in continuous time, $t_i \in \mathbb{R}^+$ (Daley & Vere-Jones, 2003). In case where the probability that an event occurs at time $t$ only depends on the past events $\mathscr{F}_t \coloneqq \{t_i, \ t_i < t\}$, PP are usually characterized through the conditional intensity function $\lambda : \mathbb{R}^+ \to \mathbb{R}^+$:

$$\lambda\left(t | \mathscr{F}_t\right) \coloneqq \lim_{\mathrm{d}t \to 0} \frac{\mathbb{P}\left(N_{t+\mathrm{d}t} - N_t = 1 | \mathscr{F}_t\right)}{\mathrm{d}t} \ , \tag{1}$$

where $N_t := \sum_{i \geq 1} \mathbf{1}_{t_i \leq t}$ is the counting process associated to the PP. This function corresponds to the expected infinitesimal rate at which events are occurring at time $t$ given the arrival times of past events prior to $t$ (Daley & Vere-Jones, 2003).

The proposed model DriPP is adapted from the Hawkes process (HP; Hawkes 1971), as the occurrence of a past event in the driver increases the likelihood of occurrence of activation events in the near future. However, here we suppose that the stochastic point process in our model of neural activations does not have the self-excitatory behavior characteristic of HP. Instead, the sources of activation in the DriPP model are either the drivers or some spontaneous background activity, but not its own previous activations. More specifically, in DriPP, the intensity function at time $t$ between a stochastic process $k$ – whose set of events is denoted $\mathcal{A}_k$ – and a non-empty set of drivers $\mathcal{P}$ – whose events are denoted $\mathcal{T}_p := \{t_1^{(p)}, \ldots, t_{n_p}^{(p)}\}, p \in \mathcal{P}$ – is composed of a baseline intensity $\mu_k \geq 0$ and triggering kernels $\kappa_{k,p} : \mathbb{R}^+ \to \mathbb{R}$:

$$\lambda_{k,\mathcal{P}}(t) = \mu_k + \sum_{p \in \mathcal{P}} \sum_{i, t_i^{(p)} \leq t} \alpha_{k,p} \kappa_{k,p}\left(t - t_i^{(p)}\right) \ , \tag{2}$$

where $\alpha_{k,p} \geq 0$ is a coefficient which controls the relative importance of the driver $p$ on the occurrence of events on the stochastic process $k$. Note that when the driver processes are known, the intensity function is deterministic, and thus corresponds to the intensity of an inhomogeneous Poisson process (Daley & Vere-Jones, 2003). The coefficient $\alpha_{k,p}$ is set to be non-negative so that we only model excitatory effects, as events on the driver only increase the likelihood of occurrence of new events on the stochastic process. Inhibition effects are assumed non-existent. Figure 1 illustrates how events $\mathcal{T}_p$ on the driver influence the intensity function after a short latency period.

A critical parametrization of this model is the choice of the triggering kernels $\kappa_{k,p}$. To best model the latency, we use a parametric truncated normal distribution of mean $m_{k,p} \in \mathbb{R}$ and standard deviation $\sigma_{k,p} > 0$, with support $[a\,;b] \subset \mathbb{R}^+$, $b > a$. Namely,

$$\kappa_{k,p}(x) := \kappa\left(x; m_{k,p}, \sigma_{k,p}, a, b\right) = \frac{1}{\sigma_{k,p}} \frac{\phi\left(\frac{x - m_{k,p}}{\sigma_{k,p}}\right)}{\Phi\left(\frac{b - m_{k,p}}{\sigma_{k,p}}\right) - \Phi\left(\frac{a - m_{k,p}}{\sigma_{k,p}}\right)} \mathbf{1}_{a \leq x \leq b} \ , \tag{3}$$

where $\phi$ (resp. $\Phi$) is the probability density function (resp. cumulative distribution function) of the standard normal distribution. This parametrization differs from the usual exponential kernel usually considered in HP, that captures responses with low latency. Note that the truncation values $a, b \in \mathbb{R}^+$ are supposed independent of both the stochastic process and the drivers, hence they are similar for all kernel $p \in \mathcal{P}$. Indeed, in the context of this paper, those values delimit the time interval during which a neuronal response might occur following an external stimulus. In other words, the interval $[a\,;b]$ denotes the range of possible latency values. In the following, we denote by $T := T^{(k)}$ the duration of the process $k$.

## 3 PARAMETERS INFERENCE WITH AN EM-BASED ALGORITHM

We propose to infer the model parameters $\Theta_{k,\mathcal{P}} = (\mu_k, \boldsymbol{\alpha}_{k,\mathcal{P}}, \boldsymbol{m}_{k,\mathcal{P}}, \boldsymbol{\sigma}_{k,\mathcal{P}})$, where we denote in bold the vector version of the parameter, *i.e.*, $\boldsymbol{x}_{k,\mathcal{P}} = (x_{k,p})_{p \in \mathcal{P}}$, via maximum-likelihood using an EM-based algorithm (Lewis & Mohler, 2011; Xu et al., 2016). The pseudocode of the algorithm is presented in Algorithm 1. The expectation-maximization (EM) algorithm (Dempster et al., 1977) is an iterative algorithm that allows to find the maximum likelihood estimates (MLE) of parameters in a probabilistic model when the latter depends on non-observable latent variables. First, from (2), we derive the negative log-likelihood of the model (see details in Appendix A.1):

$$\mathcal{L}_{k,\mathcal{P}}\left(\Theta_{k,\mathcal{P}}\right) = \mu_k T + \sum_{p \in \mathcal{P}} \alpha_{k,p} n_p - \sum_{t \in \mathcal{A}_k} \log\left(\mu_k + \sum_{p \in \mathcal{P}} \sum_{i, t_i^{(p)} \leq t} \alpha_{k,p} \kappa_{k,p}\left(t - t_i^{(p)}\right)\right) \ . \tag{4}$$

**Expectation step** For a given estimate, the E-step aims at computing the events' assignation, *i.e.*, the probability that an event comes from either the kernel or the baseline intensity. At iteration $n$,

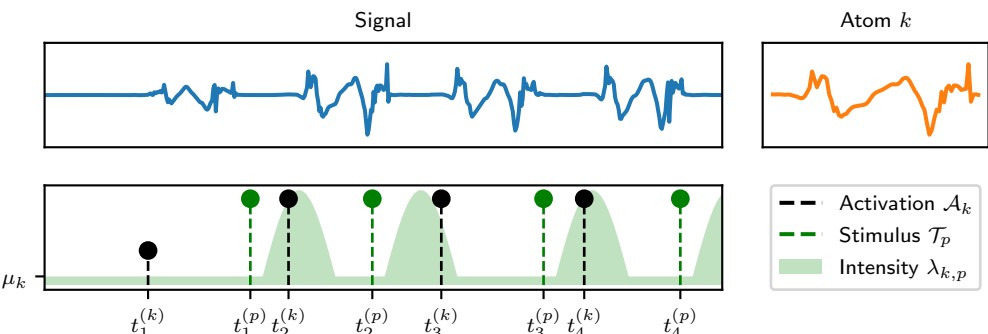

Figure 1: **Top:** Convolutional dictionary learning (CDL) applied to a univariate signal (blue) decomposes it as the convolution of a temporal pattern (orange) and a sparse activation signal (black). **Bottom:** Intensity function $\lambda_{k,p}$ defined by its baseline $\mu_k$ and the stimulus events $\mathcal{T}_p$ (green). Intensity increases following stimulation events with a certain latency.

let $P_k^{(n)}(t) \equiv P_k^{(n)}(t; p, k)$ be the probability that the activation at time $t \in [0; T]$ has been triggered by the baseline intensity of the stochastic process $k$, and $P_p^{(n)}(t) \equiv P_p^{(n)}(t; p, k)$ be the probability that the activation at time $t$ has been triggered by the driver $p$. By the definition of our intensity model (2), we have:

$$P_k^{(n)}(t) = \frac{\mu_k^{(n)}}{\lambda_{k,\mathcal{P}}^{(n)}(t)} \qquad \text{and} \qquad \forall p \in \mathcal{P}, P_p^{(n)}(t) = \frac{\alpha_{k,p}^{(n)} \sum_{i, t_i^{(p)} \leq t} \kappa_{k,p}^{(n)}\left(t - t_i^{(p)}\right)}{\lambda_{k,\mathcal{P}}^{(n)}(t)}, \quad (5)$$

where $\theta^{(n)}$ denotes the value of the parameter $\theta$ at step $n$ of the algorithm, and similarly, if $f$ is a function of parameter $\theta$, $f^{(n)}(x; \theta) := f\left(x; \theta^{(n)}\right)$. Note that $\forall t \in [0; T]$, $P_k^{(n)}(t) + \sum_{p \in \mathcal{P}} P_p^{(n)}(t) = 1$.

**Maximization step** Once this assignation has been computed, one needs to update the parameters of the model using MLE. To obtain the update equations, we fix the probabilities $P_k^{(n)}$ and $P_p^{(n)}$, and cancel the negative log-likelihood derivatives with respect to each parameter. For given values of probabilities $P_k^{(n)}(t)$ and $P_p^{(n)}(t)$, we here derive succinctly the update for parameters $\mu$ and $\boldsymbol{\alpha}$:

$$\frac{\partial \mathcal{L}_{k,\mathcal{P}}}{\partial \mu_k^{(n)}}\left(\Theta_{k,\mathcal{P}}^{(n)}\right) = 0 \Leftrightarrow T - \sum_{t \in \mathcal{A}_k} \frac{1}{\lambda_{k,\mathcal{P}}^{(n)}(t)} = 0 \Leftrightarrow T - \sum_{t \in \mathcal{A}_k} \frac{P_k^{(n)}(t)}{\mu_k^{(n)}(t)} = 0 \Leftrightarrow \mu_k^{(n+1)} = \frac{1}{T} \sum_{t \in \mathcal{A}_k} P_k^{(n)}(t) \tag{6}$$

$$\frac{\partial \mathcal{L}_{k,\mathcal{P}}}{\partial \alpha_{k,p}^{(n)}}\left(\Theta_{k,\mathcal{P}}^{(n)}\right) = 0 \Leftrightarrow n_p - \sum_{t \in \mathcal{A}_k} \frac{P_p^{(n)}(t)}{\alpha_{k,p}^{(n)}} = 0 \Leftrightarrow \alpha_{k,p}^{(n+1)} = \pi_{\mathbb{R}^+}\left(\frac{1}{n_p} \sum_{t \in \mathcal{A}_k} P_p^{(n)}(t)\right) \tag{7}$$

where $\pi_E(\cdot)$ denotes the projection onto the set $E$. These two updates amount to maximizing the probabilities that the events assigned to the driver or the baseline stay assigned to the same generation process. Then, we give the update equations for $\boldsymbol{m}$ and $\boldsymbol{\sigma}$, which corresponds to parametric estimates of each truncated Gaussian kernel parameter with events assigned to the kernel. Detailed computations are provided in Appendix A.2.

$$m_{k,p}^{(n+1)} = \frac{\alpha_{k,p}^{(n)} \sum_{t \in \mathcal{A}_k} \sum_{i, t_i^{(p)} \leq t} \frac{\left(t - t_i^{(p)}\right) \kappa_{k,p}^{(n)}\left(t - t_i^{(p)}\right)}{\lambda_{k,\mathcal{P}}^{(n)}(t)}}{\sum_{t \in \mathcal{A}_k} P_p^{(n)}(t)} - \sigma_{k,p}^{(n)2} \frac{C_m\left(m_{k,p}^{(n)}, \sigma_{k,p}^{(n)}, a, b\right)}{C\left(m_{k,p}^{(n)}, \sigma_{k,p}^{(n)}, a, b\right)} \tag{8}$$

$$\sigma_{k,p}^{(n+1)} = \pi_{[\varepsilon; +\infty)}\left(\frac{C\left(m_{k,p}^{(n)}, \sigma_{k,p}^{(n)}, a, b\right)}{C_\sigma\left(m_{k,p}^{(n)}, \sigma_{k,p}^{(n)}, a, b\right)} \frac{\alpha_{k,p}^{(n)} \sum_{t \in \mathcal{A}_k} \sum_{i, t_i^{(p)} \leq t} \frac{\left(t - t_i^{(p)}(t) - m_{k,p}^{(n)}\right)^2 \kappa_{k,p}^{(n)}\left(t - t_i^{(p)}\right)}{\lambda_{k,\mathcal{P}}^{(n)}(t)}}{\sum_{t \in \mathcal{A}_k} P_p^{(n)}(t)}\right)^{1/3} \tag{9}$$

where, $C(m, \sigma, a, b) := \int_a^b \exp\left(-\frac{1}{2}\frac{(u-m)^2}{\sigma^2}\right) du$, $C_m(m, \sigma, a, b) := \frac{\partial C}{\partial m}(m, \sigma, a, b)$ and $C_\sigma(m, \sigma, a, b) := \frac{\partial C}{\partial \sigma}(m, \sigma, a, b)$. Here $\varepsilon > 0$ is predetermined to ensure that $\sigma$ remains strictly positive. In practice, we set $\varepsilon$ such that we avoid the overfitting that can occur when the kernel's mass is too concentrated. Note that once the initial values of the parameters are determined, the EM algorithm is entirely deterministic.

---

**Algorithm 1:** EM-based algorithm

---

**input** : $\mathcal{A}_k, \mathcal{T}_p, a, b, T, N$
**output :** The estimated values for parameters
$\qquad \mu, \boldsymbol{\alpha}, \boldsymbol{m}$ and $\boldsymbol{\sigma}$

1 Initialize $\mu^{(0)}, \boldsymbol{\alpha}^{(0)}, \boldsymbol{m}^{(0)}, \boldsymbol{\sigma}^{(0)}$;
2 **for** $i = 0, \ldots, N-1$ **do**
3 $\quad$ **if** $\boldsymbol{\alpha}^{(i)} = \mathbf{0}_{\mathbb{R}^{\#\mathcal{P}}}$ **then**
4 $\quad\quad \mu^{(i+1)} = \mu^{(\text{MLE})}$;
5 $\quad\quad$ break;
6 $\quad$ **end**
7 $\quad$ Define $\lambda^{(i)}$; Compute
$\qquad \mu^{(i+1)}, \boldsymbol{\alpha}^{(i+1)}, \boldsymbol{m}^{(i+1)}, \boldsymbol{\sigma}^{(i+1)}$;
8 **end**
9 **return** $\mu^{(i+1)}, \boldsymbol{\alpha}^{(i+1)}, \boldsymbol{m}^{(i+1)}, \boldsymbol{\sigma}^{(i+1)}$

---

Also, when the estimate of parameter $m$ is too far from the kernel's support $[a\,;b]$, we are in a pathological case where EM is diverging due to indeterminacy between setting $\alpha = 0$ and pushing $m$ to infinity due to the discrete nature of our events. Thus, we consider that the stochastic process is not linked to the considered driver, and fall back to the MLE estimator. The algorithm is therefore stopped and we set $\boldsymbol{\alpha} = \mathbf{0}_{\mathbb{R}^{\#\mathcal{P}}}$.

It is worth noting that if $\forall p \in \mathcal{P}, \alpha_{k,p} = 0$, then the intensity is reduced to its baseline, thus the negative log-likelihood is $\mathcal{L}_{k,p}(\Theta_{k,p}) = \mu_k T - \#\mathcal{A}_k \log \mu_k$, where $\#\mathcal{A}$ denotes the cardinality of the set $\mathcal{A}$. Thus, we can terminate the EM algorithm by directly computing the MLE for $\mu_k$, namely: $\mu_k^{(\text{MLE})} = \#\mathcal{A}_k/T$.

**Initialization strategy** We propose a "smart start" initialization strategy, where parameters are initialized based on their role in the model. It reads:

$$\mu_k^{(0)} = \frac{\#\mathcal{A}_k - \#\left(\bigcup_{p \in \mathcal{P}} \mathcal{D}_{k,p}\right)}{T - \boldsymbol{\lambda}\left(\bigcup_{p \in \mathcal{P}} \bigcup_{t' \in \mathcal{T}_p} [t' + a\,;t' + b]\right)} \tag{10}$$

$$\alpha_{k,p}^{(0)} = \frac{\#\mathcal{D}_{k,p}}{\boldsymbol{\lambda}\left(\bigcup_{t' \in \mathcal{T}_p} [t' + a\,;t' + b]\right)} - \mu_k^{(0)}, \quad \forall p \in \mathcal{P} \tag{11}$$

$$m_{k,p}^{(0)} = \frac{1}{\#\mathcal{D}_{k,p}} \sum_{d \in \mathcal{D}_{k,p}} d \quad \text{and} \quad \sigma_{k,p}^{(0)} = \sqrt{\frac{1}{\#\mathcal{D}_{k,p}} \sum_{d \in \mathcal{D}_{k,p}} \left|d - m^{(0)}\right|^2}, \quad \forall p \in \mathcal{P}, \tag{12}$$

where $\boldsymbol{\lambda}(\cdot)$ denotes the Lebesgue measure, and where $\mathcal{D}_{k,p} := \{t - t_*^{(p)}(t), \ t \in \mathcal{A}_k\} \cap [a\,;b]$ is the set of all empirical delays possibly linked to the driver $p$, with $t_*^{(p)}(t) := \max\{t', \ t' \in \mathcal{T}_p, t' \leq t\}$ denoting the timestamp of the last event on the driver $p$ that occurred before time $t$. Here, the baseline intensity $\mu^{(0)}$ is set to the average number of process' events that occur outside any kernel support, *i.e.,* the events that are guaranteed to be exogenous or spontaneous. Similarly, the kernel intensity $\alpha^{(0)}$ is computed as the increase in the average number of activations over the kernel support, compared to $\mu^{(0)}$. The initial guess for $m^{(0)}$ and $\sigma^{(0)}$ are obtained with their parametric estimates, considering that all event on the kernel support are assigned to the considered driver.

## 4 EXPERIMENTS

We evaluated our model on several experiments, using both synthetic and empirical MEG data. We used Python (Python Software Foundation, 2019) and its scientific libraries (Virtanen et al., 2020; Hunter, 2007; Harris et al., 2020). We relied on alphacsc for CDL with rank-1 constraints on MEG (Dupré la Tour et al., 2018) and we used MNE (Gramfort et al., 2013) to load and manipulate the MEG datasets. Computations were run on CPU Intel(R) Xeon(R) E5-2699, with 44 physical cores.

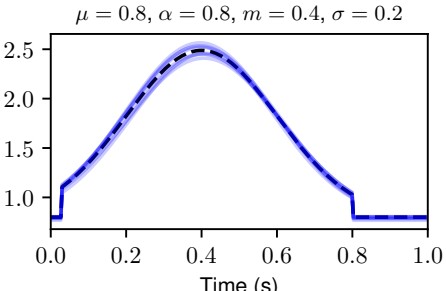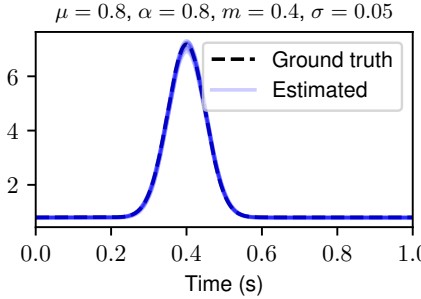

Figure 2: True and estimated intensity functions following a driving event at time zero for two different kernels, on synthetic data. **Left**: "wide" kernel with $\sigma = 0.2$. **Right**: "sharp" kernel with $\sigma = 0.05$. Parameters used are $T = 10000$, $P/S = 0.6$. On synthetic data, the EM algorithm successfully retrieves the true values of parameters, for both shapes of kernels.

### 4.1 EVALUATION OF THE EM CONVERGENCE ON SYNTHETIC DATA

For a given number of drivers and a set of corresponding parameters $\Theta$, we first generate the drivers' processes and then simulate the stochastic process for a pre-determined duration $T$. Each driver's timestamps are simulated as follows: given an interstimuli interval (ISI), a set of $S = \left\lfloor \frac{T}{\text{ISI}} \right\rfloor$ equidistant timestamps is generated – where $\lfloor \cdot \rfloor$ denotes the floor function. Then $P$ timestamps are uniformly sampled without replacement from this set. In all our experiments, we fixed the ISI to $1\,\text{s}$ for the "wide" kernel, and to $1.4\,\text{s}$ for the "sharp" one. Finally, a one-dimensional non-homogeneous Poisson process is simulated following Lewis' thinning algorithm (Lewis & Shedler, 1979), given the predefined intensity function $\lambda$ and the drivers' timestamps.

Figure 2 illustrates the intensity function recovery with two drivers considered together: the first one has a "wide" kernel with standard deviation $\sigma = 0.2\,\text{s}$, and the second one has a "sharp" kernel with $\sigma = 0.05\,\text{s}$. Both kernels have support $[0.03\,\text{s}\,;0.8\,\text{s}]$ and mean $m = 0.4\,\text{s}$, the coefficients $\alpha$ are both set to 0.8 and the baseline intensity parameter $\mu$ to 0.8. We report 8 estimated intensities obtained from independent simulations of the processes – using $T = 10000\,\text{s}$ and $P/S = 0.6$ – that we plot over each one of the driver's kernel's support. The EM algorithm is run for 50 iterations using the "smart start" initialization strategy described in Section 2. Note that here, the randomness only comes from the data generation, as the EM algorithm uses a deterministic initialization. Figures demonstrate that the EM algorithm is able to successfully recover the parameters for both shapes of kernels.

To provide a quantitative evaluation of the parameters' recovery, we compute, for each driver $p \in \mathcal{P}$, the $\ell_\infty$ norm between the intensity $\lambda^*$ computed with the true parameters $\Theta_p^*$ and the estimated intensity $\lambda_p$ with parameters $\widehat{\Theta}_p$:

$$\left\| \lambda^* - \widehat{\lambda}_p \right\|_\infty := \max_{t \in [0\,;T]} \left| \mu^* + \alpha_p^* \kappa_p^*(t) - \hat{\mu} - \hat{\alpha}_p \hat{\kappa}_p(t) \right| \quad . \tag{13}$$

The rationale for using the $\ell_\infty$ norm is to ensure that errors during baseline and within the kernel support are given equal importance. Figure 3 presents the parameter recovery for the same scenario with varying $P/S$ and $T$. To easily compare the EM performances on the two shapes of kernels, Figure 3 (resp. Figure A.1) reports the mean (resp. standard deviation) of the relative $\ell_\infty$ norm – that is the $\ell_\infty$ divided by the maximum of the true intensity $\lambda^*$ – computed for each of the driver over 30 repetitions with different realizations of the process. The results show that the more data are available, either due to a longer process duration (increase in $T$) or due to a higher event density (increase in $P/S$), the better are the parameter estimates. The convergence appears to be almost linear in these two cases. Moreover, the average computation time for an EM algorithm in Figure 3 took $18.16\,\text{s}$, showing the efficiency of our inference method. In addition, we report in appendix the scaling of the EM computation time as a function of $T$ in Figure A.2.

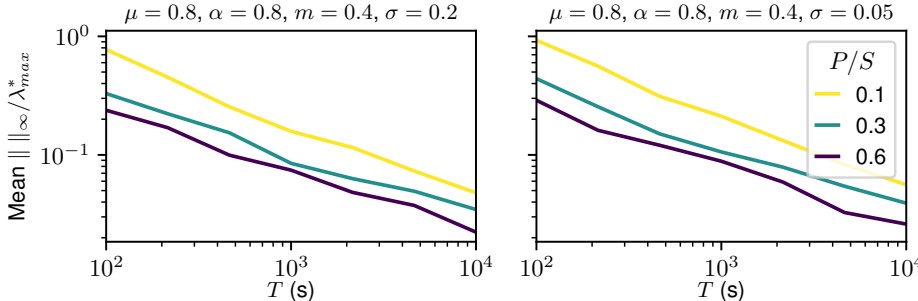

Figure 3: Mean of the relative infinite norm as a function of process duration $T$ and the percentage of events kept $P/S$, for two kernel shapes on synthetic data: wide kernel (**left**) and sharp kernel (**right**). The accuracy of the EM estimates increases with longer and denser processes.

## 4.2 EVOKED AND INDUCED EFFECTS CHARACTERIZATION IN MEG DATA

**Datasets** Experiments on MEG data were run on two datasets from MNE Python package (Gramfort et al., 2014; 2013): the *sample* dataset and the somatosensory (*somato*) dataset[1]. These datasets were selected as they elicit two distinct types of event-related neural activations: evoked responses which are time locked to the onset of the driver process, and induced responses which exhibit random jitters. Complementary experiments were performed on the larger Cam-CAN dataset (Shafto et al., 2014)[2]. Presentation of the dataset, data pre-processing and obtained results on 3 subjects are presented in Appendix A.7. The presented results are self-determined as they exhibit, for each subject, the atoms that have the higher ratio $\alpha/\mu$. For all studied datasets, full results are presented in supplementary materials.

The *sample* dataset contains M/EEG recordings of a human subject presented with audio and visual stimuli. In this experiment, checkerboard patterns are presented to the subject in the left and right visual field, interspersed by tones to the left or right ear. The experiment lasts about $4.6\,\mathrm{min}$ and approximately 70 stimuli per type are presented to the subject. The interval between the stimuli is on average of $750\,\mathrm{ms}$, all types combined, with a minimum of $593\,\mathrm{ms}$. Occasionally, a smiley face is presented at the center of the visual field. The subject was asked to press a button with the right index finger as soon as possible after the appearance of the face. In the following, we are only interested in the four main stimuli types: auditory left, auditory right, visual left, and visual right. For the *somato* dataset, a human subject is scanned with MEG during $15\,\mathrm{min}$, while 111 stimulations of his left median nerve were made. The minimum ISI is $7\,\mathrm{s}$.

**Experimental setting** For both datasets, only the 204 gradiometer channels are analyzed. The signals are pre-processed using high-pass filtering at $2\,\mathrm{Hz}$ to remove slow drifts in the data, and are resampled to $150\,\mathrm{Hz}$ to limit the atom size in the CDL. CDL is computed using alphacsc (Dupré la Tour et al., 2018) with the GreedyCDL method. For the *sample* dataset, 40 atoms of duration $1\,\mathrm{s}$ each are extracted, and for the *somato* dataset, 20 atoms of duration $0.53\,\mathrm{s}$ are estimated. The extracted atoms' activations are binarized using a threshold of $6 \times 10^{-11}$ (resp. $1 \times 10^{-10}$) for *sample* (resp. *somato*), and the times of the events are shifted to make them correspond to the peak amplitude time of the atom. Then, for every atom, the intensity function is estimated using the EM-based algorithm with 400 iterations and the "smart start" initialization strategy. Kernels' truncation values are hyper-parameters for the EM and thus must be pre-determined. The upper truncation value $b$ is chosen smaller than the minimum ISI. Here, we used in addition some previous domain knowledge to set coherent values for each dataset. Hence, for the *sample* (resp. *somato*) dataset, kernel support is fixed at $[0.03\,\mathrm{s}\,;0.5\,\mathrm{s}]$ (resp. $[0\,\mathrm{s}\,;2\,\mathrm{s}]$). See Appendix A.4 for an analysis on how these hyperparameters influence on the obtained results presented below.

**Evoked responses in sample dataset** Results on the *sample* dataset are presented in Figure 4. We plot the spatial and temporal representations of four selected atoms, as well as the estimated intensity functions related to the two types of stimuli: auditory (*blue*) and visual (*orange*). The first two atoms

---

[1]Both available at https://mne.tools/stable/overview/datasets_index.html
[2]Available at https://www.cam-can.org/index.php?content=dataset

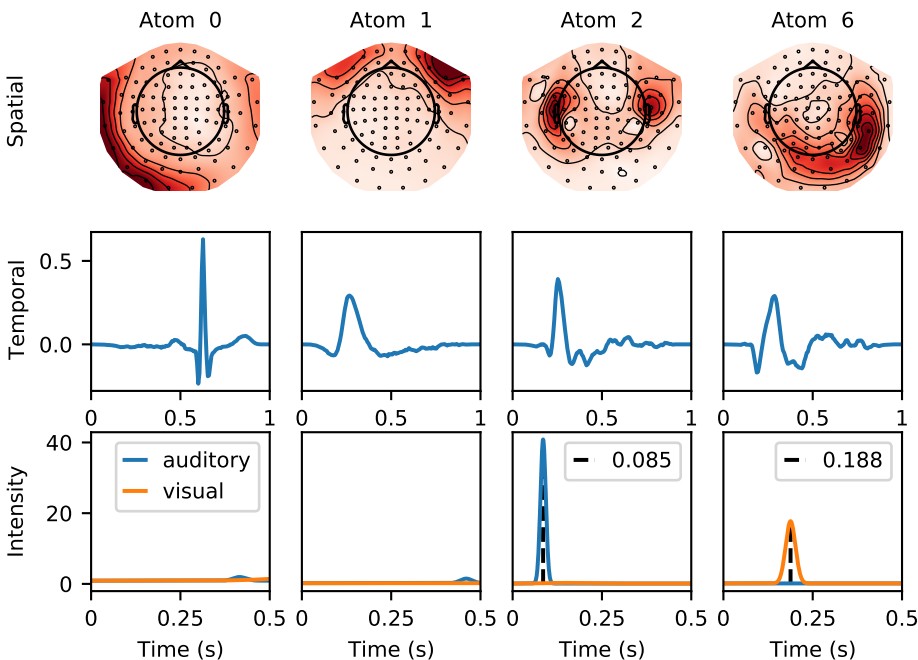

Figure 4: Spatial and temporal patterns of 4 atoms from *sample* dataset, and their respective estimated intensity functions following a stimulus (cue at time = 0 s), for auditory and visual stimuli. The heartbeat and eye-blink artifacts are not linked to any stimuli. An auditory stimulus will induce a neural response similar to atom 2, with a mean latency of $85\,\mathrm{ms}$.

are specifically handpicked to exhibit the usual artifacts, and the last two are selected as they have the two bigger ratios $\alpha/\mu$ for their respective learned intensity functions. Even though the intensity is learned with the two stimuli conjointly, we plot the two corresponding "intensities at the kernel separately, *i.e.,* $\forall p \in \mathcal{P}, \forall t \in [0\,;0.5]$, we plot $\lambda_{k,p}(t), k = 0, 1, 2, 6$.

Spatial and temporal representations of atom 0 (resp. atom 1) indicate that it corresponds to the heartbeat (resp. the eye blink) artifact. These two atoms are thus expected not to be linked to any stimuli. This is confirmed by the shape of the intensities estimated with DriPP that is mostly flat, which indicates that the activation of these two atoms are independent of auditory and visual stimuli. Note that these two artifacts can also be recovered by an Independent Component Analysis (ICA), as shown in Figure A.5. Indeed, the cosine similarity between the spatial maps of the eye blink (resp. the heartbeat) artifact extracted with CDL and its corresponding component in ICA analysis is $99.58\,\%$ (resp. $99.78\,\%$), as presented in Figure A.6. In contrast, by looking at the spatial and temporal patterns of atom 2 (resp. atom 6), it can be associated with an auditory (resp. visual) evoked response. Given the spatial topography of atom 2, we conclude to a bilateral auditory response and the peak transient temporal pattern suggests an evoked response that is confirmed by the estimated intensity function that contains a narrow peak around $85\,\mathrm{ms}$ post-stimulus. This is the M100 response – here the auditory one – well known in the MEG literature (its equivalent in EEG is the N100) (Näätänen & Picton, 1987). The M100 is indeed a peak observed in the evoked response between 80 and 120 milliseconds after the onset of a stimulus in an adult population. Regarding atom 6, topography is right lateralized in the occipital region, suggesting a visual evoked response. This is confirmed by the intensity function estimated that reports a relationship between this atom and the visual stimuli. Here also, the intensity peak is narrow, which is characteristic of an evoked response. This reflects a right lateralized response along the right ventral visual stream in this subject. This may be connected to the P200, a peak of the electrical potential between 150 and 275 ms after a visual onset. Moreover, the intensities estimated with DriPP for the unrelated tasks are completely flat. We have $\alpha = 0$, which indicates that atoms' activations are exogenous or spontaneous relatively to unrelated stimuli. For comparison, we present in Appendix A.5 similar results obtained with dedicated MEG data analysis tools, such as evoked responses and time-frequency plots.

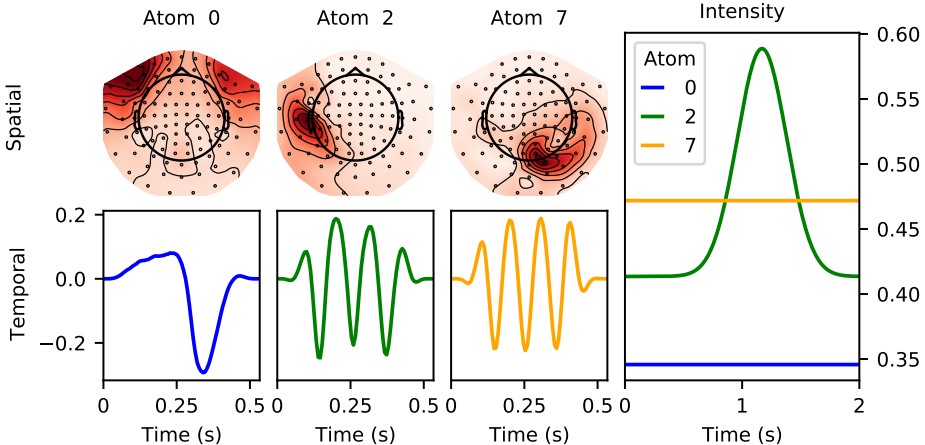

Figure 5: Spatial and temporal patterns of 3 atoms from *somato* dataset, and their respective estimated intensity functions following a somatosensory stimulus (cue at time = 0 s). The eye-blink artifact (atom 0) is not linked to the stimulus, and neither is the $\alpha$-wave (atom 7). A somatosensory stimulus will induce a neural response similar to atom 2, with a mean latency of $1$ s.

**Induced response in somato dataset**  Results on the *somato* dataset are presented in Figure 5. Similar to the results on *sample*, spatial and temporal patterns of 3 handpicked atoms are plotted alongside the intensity functions obtained with DriPP. Thanks to their spatial and temporal patterns, and with some domain knowledge, it is possible to categorize these 3 atoms: atom 2 corresponds to a $\mu$-wave located in the secondary somatosensory region (S2), atom 7 corresponds to an $\alpha$-wave originating in the occipital visual areas, whereas atom 0 corresponds to the eye-blink artifact. As $\alpha$-waves are spontaneous brain activity, they are not phase-locked to the stimuli. It is thus expected that atom 7 is not linked to the task, as confirmed by its estimated intensity function where $\alpha = 0$. For atom 2 – that corresponds to a $\mu$-wave –, its respective intensity is nonflat with a broad peak close to $1$ s, which characterizes an induced response. Moreover, similar to results on the *sample* dataset, we recover the eye-blink artifact that also has a flat intensity function. This allows us to be confident in the interpretation of the obtained results. Some other $\mu$-wave atoms – atoms 1 and 4 – are presented in Figure A.10 in Appendix A.6. They have an estimated intensity similar to atom 2, *i.e.,* non-flat with a broad peak close to $1$ s. The usual time/frequency analysis reported in Figure A.9 exhibits the induced response of the $\mu$-wave.

## 5   DISCUSSION

This work proposed a point process (PP) based approach specially designed to model how external stimuli can influence the occurrences of recurring spatio-temporal patterns, called *atoms*, extracted from M/EEG recordings using convolutional dictionary learning (CDL). The key advantage of the developed method is that by estimating few parameters (one baseline parameter and 3 parameters per considered driver), it provides a direct statistical characterization of when and how each stimulus is responsible for the occurrences of neural responses. Importantly, it can achieve this with relatively limited data which is well adapted to MEG/EEG experiments that last only a few minutes, hence leading to tens or hundreds of events at most. This work proposed an EM algorithm derived for a novel kernel function: the truncated Gaussian, which differs from the usual parametrization in PP models that capture immediate responses, *e.g.,* with exponential kernels. As opposed to competing methods that can involve manual selection of task-related neural sources, DriPP offers a unified approach to extract waveforms and automatically select the ones that are likely to be triggered by the considered stimuli. Note however that DriPP has been developed based on a point process framework, which is event-based. When working with continuous stimuli, other techniques must be considered (*e.g.,* STRF, spatio-temporal response functions; Drennan & Lalor 2019). Future work will explore the modeling of inhibitions effects ($\alpha < 0$) frequently observed in neuroscience data. Yet, this could potentially degrade the interpretability of the model parameters, and it also requires a new definition of the intensity function as one needs to ensure it stays non-negative at all times.

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

## A  APPENDIX

### A.1  DETAILS OF THE NEGATIVE LOG-LIKELIHOOD COMPUTATION

We defined the negative log-likelihood for parameters $\Theta_{k,\mathcal{P}} = (\mu_k, \boldsymbol{\alpha}_{k,\mathcal{P}}, \boldsymbol{m}_{k,\mathcal{P}}, \boldsymbol{\sigma}_{k,\mathcal{P}})$ as follows:

$$\mathcal{L}_{k,\mathcal{P}}(\Theta_{k,\mathcal{P}}) = \int_0^T \lambda_{k,\mathcal{P}}(t)\mathrm{d}t - \sum_{t\in\mathcal{A}_k} \log \lambda_{k,\mathcal{P}}(t) \ . \tag{14}$$

We show here in details that $\int_0^T \lambda_{k,p}(t)\mathrm{d}t = \mu_k T + \sum_{p\in\mathcal{P}} \alpha_{k,p} n_p$. Without loss of generality, we can first assume that $\forall p \in \mathcal{P}, t_{n_p}^{(p)} + b \leq T$. Hence,

$$\forall p \in \mathcal{P}, \forall i = 1, \ldots, n_p, \int_0^T \kappa_{k,p}\left(t - t_i^{(p)}\right) \mathrm{d}t = 1 \ .$$

Thus, we have that

$$\int_0^T \lambda_{k,p}(t)\,\mathrm{d}t = \int_0^T \left(\mu_k + \sum_{p\in\mathcal{P}} \sum_{i, t_i^{(p)}<t} \alpha_{k,p}\kappa_{k,p}\left(t - t_i^{(p)}\right)\right) \mathrm{d}t$$

$$= \mu_k T + \sum_{p\in\mathcal{P}} \alpha_{k,p} \left(\sum_{t_i^{(p)}\in\mathcal{T}_p} \int_0^T \kappa_{k,p}\left(t - t_i^{(p)}\right) \mathrm{d}t\right)$$

$$= \mu_k T + \sum_{p\in\mathcal{P}} \alpha_{k,p} n_p \ .$$

Finally, we have that the negative log-likelihood writes

$$\mathcal{L}_{k,\mathcal{P}}(\Theta_{k,\mathcal{P}}) = \mu_k T + \sum_{p\in\mathcal{P}} \alpha_{k,p} n_p - \sum_{t\in\mathcal{A}_k} \log \lambda_{k,\mathcal{P}}(t) \ .$$

### A.2  DETAILS OF EM-BASED ALGORITHM COMPUTATIONS

First, recall that the negative log-likelihood for parameters $\Theta_{k,\mathcal{P}} = (\mu_k, \boldsymbol{\alpha}_{k,\mathcal{P}}, \boldsymbol{m}_{k,\mathcal{P}}, \boldsymbol{\sigma}_{k,\mathcal{P}})$ is as follows:

$$\mathcal{L}_{k,\mathcal{P}}(\Theta_{k,\mathcal{P}}) = \mu_k T + \sum_{p\in\mathcal{P}} \alpha_{k,p} n_p - \sum_{t\in\mathcal{A}_k} \log \left(\mu_k + \sum_{p\in\mathcal{P}} \sum_{i, t_i^{(p)}\leq t} \alpha_{k,p}\kappa_{k,p}\left(t - t_i^{(p)}; m_{k,p}, \sigma_{k,p}\right)\right) \ .$$

Recall also that we defined $P_k^{(n)}$ and $P_p^{(n)}$ as follows:

$$P_k^{(n)}(t) = \frac{\mu_k^{(n)}}{\lambda_{k,\mathcal{P}}^{(n)}(t)} \qquad \text{and} \qquad \forall p \in \mathcal{P}, P_p^{(n)}(t) = \frac{\alpha_{k,p}^{(n)} \sum_{i, t_i^{(p)}\leq t} \kappa_{k,p}^{(n)}\left(t - t_i^{(p)}(t)\right)}{\lambda_{k,\mathcal{P}}^{(n)}(t)} \ , \tag{15}$$

where $\theta^{(n)}$ denotes the value of the parameter $\theta$ at step $n$ of the algorithm, and similarly, if $f$ is a function of parameter $\theta$, $f^{(n)}(x; \theta) := f(x; \theta^{(n)})$.

Below are details of calculation for parameters update for $m$ and $\sigma$. We first rewrite the kernel function to have it under a form that simplifies further computations.

$$\kappa(x; m, \sigma, a, b) = \frac{1}{\sigma} \frac{\phi\left(\frac{x-m}{\sigma}\right)}{\Phi\left(\frac{b-m}{\sigma}\right) - \Phi\left(\frac{a-m}{\sigma}\right)} \mathbf{1}_{a\leq x\leq b}$$

$$= \frac{\exp\left(-\frac{1}{2}\frac{(x-m)^2}{\sigma^2}\right)}{C(m, \sigma, a, b)} \mathbf{1}_{a\leq x\leq b}$$

where

$$C\left(m, \sigma, a, b\right) := \int_a^b \exp\left(-\frac{1}{2}\frac{(u-m)^2}{\sigma^2}\right) \mathrm{d}u \ . \tag{16}$$

Hence, we can precompute its derivatives with respect to $m$ and $\sigma$:

$$\frac{\partial}{\partial m}\kappa(x; m, \sigma, a, b) = \left(\frac{x-m}{\sigma^2} - \frac{C_m\left(m, \sigma, a, b\right)}{C\left(m, \sigma, a, b\right)}\right)\kappa(x; m, \sigma, a, b) \ , \tag{17}$$

and

$$\frac{\partial}{\partial \sigma}\kappa(x; m, \sigma, a, b) = \left(\frac{(x-m)^2}{\sigma^3} - \frac{C_\sigma\left(m, \sigma, a, b\right)}{C\left(m, \sigma, a, b\right)}\right)\kappa(x; m, \sigma, a, b) \ . \tag{18}$$

where $C_m\left(m, \sigma, a, b\right) := \frac{\partial C}{\partial m}\left(m, \sigma, a, b\right)$ and $C_\sigma\left(m, \sigma, a, b\right) := \frac{\partial C}{\partial \sigma}\left(m, \sigma, a, b\right)$.

**Update equation for $m_{k,p}$**

$$\frac{\partial \mathcal{L}_{k,\mathcal{P}}}{\partial m_{k,p}}\left(\Theta_{k,\mathcal{P}}\right) = -\sum_{t\in\mathcal{A}_k}\sum_{i,t_i^{(p)}\leq t}\left(\frac{t-t_i^{(p)}-m_{k,p}}{\sigma_{k,p}^2} - \frac{C_m\left(m_{k,p}, \sigma_{k,p}, a, b\right)}{C\left(m_{k,p}, \sigma_{k,p}, a, b\right)}\right)\frac{\alpha_{k,p}\kappa_{k,p}\left(t-t_i^{(p)}\right)}{\lambda_{k,\mathcal{P}}(t)}$$

$$= \left(\frac{m_{k,p}}{\sigma_{k,p}^2} + \frac{C_m\left(m_{k,p}, \sigma_{k,p}, a, b\right)}{C\left(m_{k,p}, \sigma_{k,p}, a, b\right)}\right)\sum_{t\in\mathcal{A}_k}P_p(t) - \frac{\alpha_{k,p}}{\sigma_{k,p}^2}\sum_{t\in\mathcal{A}_k}\sum_{i,t_i^{(p)}\leq t}\frac{\left(t-t_i^{(p)}\right)\kappa_{k,p}\left(t-t_i^{(p)}\right)}{\lambda_{k,\mathcal{P}}(t)} \tag{19}$$

Hence, by canceling the previous derivative,

$$m_{k,p}^{(n+1)} = \frac{\alpha_{k,p}^{(n)}\sum_{t\in\mathcal{A}_k}\sum_{i,t_i^{(p)}\leq t}\frac{\left(t-t_i^{(p)}\right)\kappa_{k,p}^{(n)}\left(t-t_i^{(p)}\right)}{\lambda_{k,\mathcal{P}}^{(n)}(t)}}{\sum_{t\in\mathcal{A}_k}P_p^{(n)}(t)} - \sigma_{k,p}^{(n)2}\frac{C_m\left(m_{k,p}^{(n)}, \sigma_{k,p}^{(n)}, a, b\right)}{C\left(m_{k,p}^{(n)}, \sigma_{k,p}^{(n)}, a, b\right)} \tag{20}$$

**Update equation for $\sigma_{k,p}$**

$$\frac{\partial \mathcal{L}_{k,\mathcal{P}}}{\partial \sigma_{k,p}}\left(\Theta_{k,\mathcal{P}}\right) = -\sum_{t\in\mathcal{A}_k}\sum_{i,t_i^{(p)}\leq t}\left(\frac{\left(t-t_i^{(p)}-m_{k,p}\right)^2}{\sigma_{k,p}^3} - \frac{C_\sigma\left(m_{k,p}, \sigma_{k,p}, a, b\right)}{C\left(m_{k,p}, \sigma_{k,p}, a, b\right)}\right)\frac{\alpha_{k,p}\kappa_{k,p}\left(t-t_i^{(p)}\right)}{\lambda_{k,\mathcal{P}}(t)}$$

$$= \frac{C_\sigma\left(m_{k,p}, \sigma_{k,p}, a, b\right)}{C\left(m_{k,p}, \sigma_{k,p}, a, b\right)}\sum_{t\in\mathcal{A}_k}P_p(t) - \frac{\alpha_{k,p}}{\sigma_{k,p}^3}\sum_{t\in\mathcal{A}_k}\sum_{i,t_i^{(p)}\leq t}\frac{\left(t-t_i^{(p)}(t)-m_{k,p}\right)^2\kappa_{k,p}\left(t-t_i^{(p)}\right)}{\lambda_{k,\mathcal{P}}(t)} \tag{21}$$

Hence, by canceling the previous derivative,

$$\sigma_{k,p}^{(n+1)} = \left(\frac{C\left(m_{k,p}^{(n)}, \sigma_{k,p}^{(n)}, a, b\right)}{C_\sigma\left(m_{k,p}^{(n)}, \sigma_{k,p}^{(n)}, a, b\right)}\frac{\alpha_{k,p}^{(n)}\sum_{t\in\mathcal{A}_k}\sum_{i,t_i^{(p)}\leq t}\frac{\left(t-t_i^{(p)}(t)-m_{k,p}^{(n)}\right)^2\kappa_{k,p}^{(n)}\left(t-t_i^{(p)}\right)}{\lambda_{k,\mathcal{P}}^{(n)}(t)}}{\sum_{t\in\mathcal{A}_k}P_p^{(n)}(t)}\right)^{1/3} \tag{22}$$

Finally, to ensure that the $\sigma$ coefficient stays strictly positive in order to avoid computational errors, we add a projection step onto $[\varepsilon\,;+\infty)$, with $\varepsilon > 0$: $\sigma_{k,p}^{(n+1)} = \pi_{[\varepsilon\,;+\infty)}\left(\sigma_{k,p}^{(n+1)}\right)$.

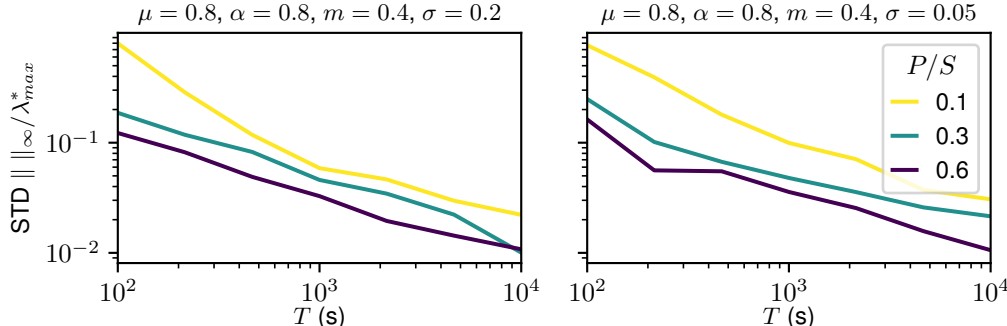

Figure A.1: Standard deviation of the relative infinite norm as a function of the process duration $T$ and the percentage of events kept $P/S$, for two kernel shapes on synthetic data: wide kernel (**left**) and sharp kernel (**right**). The variance of the EM estimates continually decreases with longer and denser processes.

### A.3 EXPERIMENTS ON SYNTHETIC DATA

In this section, we present figures that are complementary to the ones obtained on synthetic data, presented in subsection 4.1. In Figure A.1, we report the standard deviation associated with Figure 3, *i.e.,* the standard deviation of the relative infinite norm for different process duration $T$ and different proportion of events kept $P/S$, for the two kernel shapes.

Figure A.2 reports some details on the computation time of the experiment done on synthetic data which produced the Figure 3 and Figure A.1. For each value of $T$, the mean computation time is computed over 90 experiments (3 values of $P/S$ times 30 random seeds). Results are presented in Figure A.2.

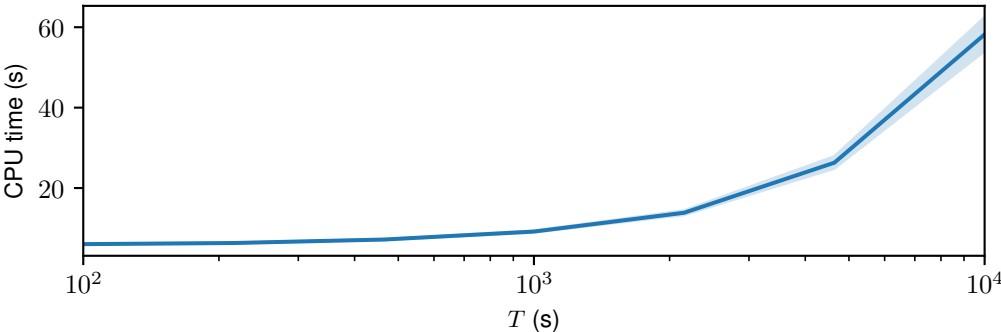

Figure A.2: Mean and $95\,\%$ CI of computation time (in seconds) for one EM algorithm, as a function of the process duration $T$ (in seconds). Results are obtained on synthetic data.

### A.4 IMPACT OF MODEL HYPERPARAMETER

In this section, we dwell on the analysis of how setting hyperparameter values may impact the obtained results, with the aim of determining whether it is possible to set these parameters using a general rule of thumb, without degrading previous results. More specifically, we will look at the impact of two hyperparameters on the results we obtained on the MNA sample dataset: the threshold value – applied to the atoms' activation values to binarized them, currently set at $6 \times 10^{-11}$ –, and the kernel support, currently set at $[0.03\,\text{s}\,;0.5\,\text{s}]$ using previous and domain knowledge. To do so, we conducted two experiments on *sample* where each varies one hyperparameter. We plot the intensity function learned by DriPP for the same four atoms – namely, the two artifacts (heartbeat and eye

blink) and the auditory and visual responses –, separately for the two stimuli (auditory and visual), similarly to Figure 4.

For the first experiment, presented in Figure A.3, we varied the threshold, expressed as a percentile, between 0 and 80. A threshold of 20 means that we only keep activations whose values are above the $20\%$ percentile computed over all strictly positive activations, *i.e.,* the smaller the threshold is, the more activations are kept. The value of the threshold used in all other experiments is the $60\%$ percentile. One can observe that for the two artifacts (atoms 0 and 1), when the threshold gets smaller, the learned intensity functions get flatter, indicating that the stimulus has no influence on the atom activation. However, for the two others atoms, the effect of a smaller threshold is the opposite, as the intensity functions have a higher peak, indicating a bigger value of the $\alpha$ parameter, and thus strengthening the link stimulus-atom. Thus, the threshold value could be set to a small percentile and therefore computed without manual intervention, without degrading the current results.

For the second experiment, we now focus on the kernel truncation values $a$ and $b$. Results are presented in Figure A.4. We set $a = 0$, as we did for somato and Cam-CAN datasets, and we vary $b$ between $0.5$ – the current value – and $10$, a large value compared to the ISI. One can observe that for $b = 0.5$ or $b = 1$, the results are either unchanged (atoms 1, 2, and 6) or better (atom 0, as the intensity function is totally flat for this artifact), indicating that setting $a = 0$ and $b$ close to the average ISI of $0.750\,\text{s}$ does not hinder the results. However, when $b$ is too large, the results degrade quickly, up to the point where all learned intensities are flat lines, indicating that our model does not find any link between the stimuli and the atoms. This is due to the fact that this hyperparameter is of great importance in the initialization step, as the greater $b$ is, the more atom's activations are considered being on a kernel support. Thus, setting the upper truncation value to a value close to the average ISI seems to give reliable results.

## A.5 Usual M/EEG data analysis

We present in this section some results obtained using usual M/EEG data analysis, such as Independent Component Analysis (ICA), epoch averaging, or time/frequency analysis. First, on MNE *sample* dataset, we proceed to an ICA to manually identify usual artifacts. To do so, similarly as the CDL pre-processing, the raw signals are filtered (high-pass filter at 2 Hz), and 40 independent components are fitted. The two components 1 and 3, that we manually identify as corresponding to the eye blink and heartbeat artifacts, are presented in Figure A.5. In Figure A.6, we associate for each of the CDL atoms presented in Figure 4 the ICA component that has the maximum cosine similarity. One can observe that the artifact atoms and components are highly similar, suggesting that CDL and ICA have equal performance on the artifact detection. Note that this high similarity is only based on the spatial pattern. Indeed, ICA does not provide temporal waveforms for the atoms as well as their temporal onsets, contrarily to CDL.

However, for the auditory and visual response, the result is different. For the auditory one (atom 1), there is not really an ICA equivalent, as it is most correlated with the eye-blink ICA component. Regarding the visual atom (atom 6), there is an ICA component that presents a high similarity. While the two related components correspond to neural sources on the occipital cortex, the atom 6 obtained with CSC is more right lateralized, suggesting a source in the right ventral visual stream. Note that unlike ICA, which recover full time courses for each source, CDL also provides the onset of the patterns, which we later use for automated identification of event related components. Finally, this demonstrates that CDL is a strong competitor to ICA for artifact identification, while simultaneously enabling to reveal evoked or induced neural responses in an automated way.

As mentioned, the ICA is commonly used to remove manually identified artifacts to reconstruct the original signals free of those artifacts. However, there are two drawbacks of this method, the first one being that some domain-related knowledge is needed in order to correctly identify the artifacts. The second drawbacks happen when the signal is reconstructed after the removal of certain components. Indeed, such a reconstruction will lead to a loss of information across all channels, as the artifacts are shared by all the sensors. Thanks to the Convolutional dictionary learning (CDL) that extracts the different artifacts directly from the raw data, our method does not suffer from these drawbacks.

Still on MNE *sample* dataset, we compute from raw data the epoch average following an auditory stimulus (stimuli in the left ear and in the right ear are combined) and plot on Figure A.7 the obtained

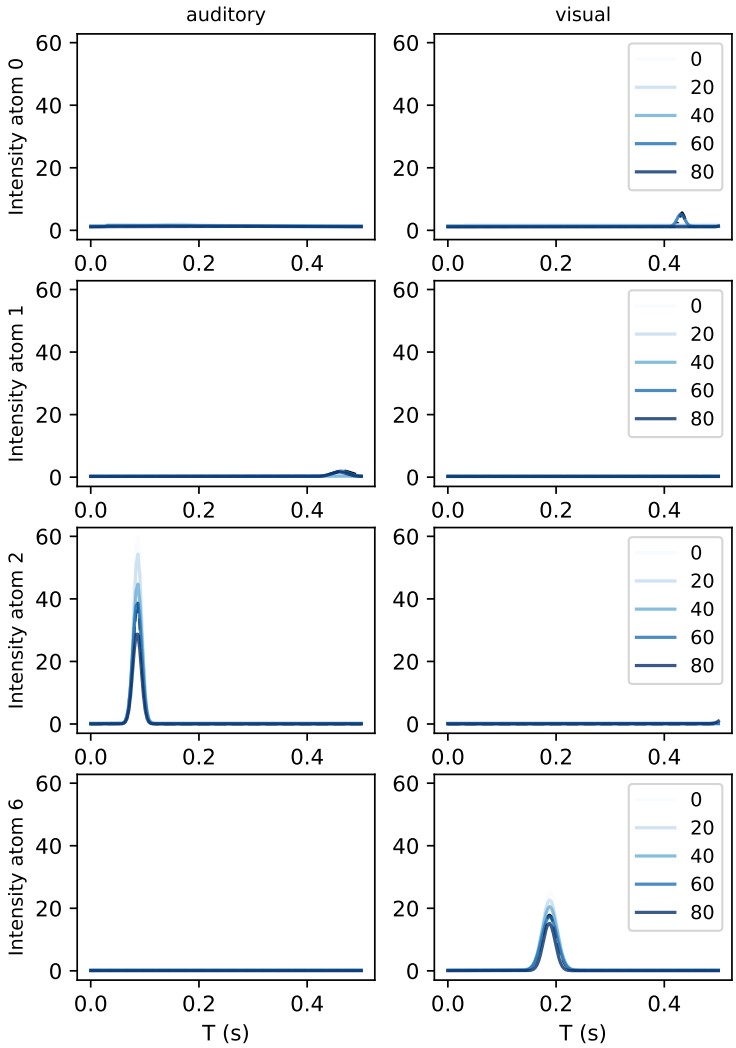

Figure A.3: Influence of the threshold (expressed as a percentile) on the obtained results on MNE sample dataset, for the 4 main atoms, for auditory and visual stimulus. The value of the threshold presented in Figure 4 is $\tau = 60\,\%$. The threshold value has limited impact on obtained results, and thus could be determined with a general rule of thumb, as a percentile over all activations values.

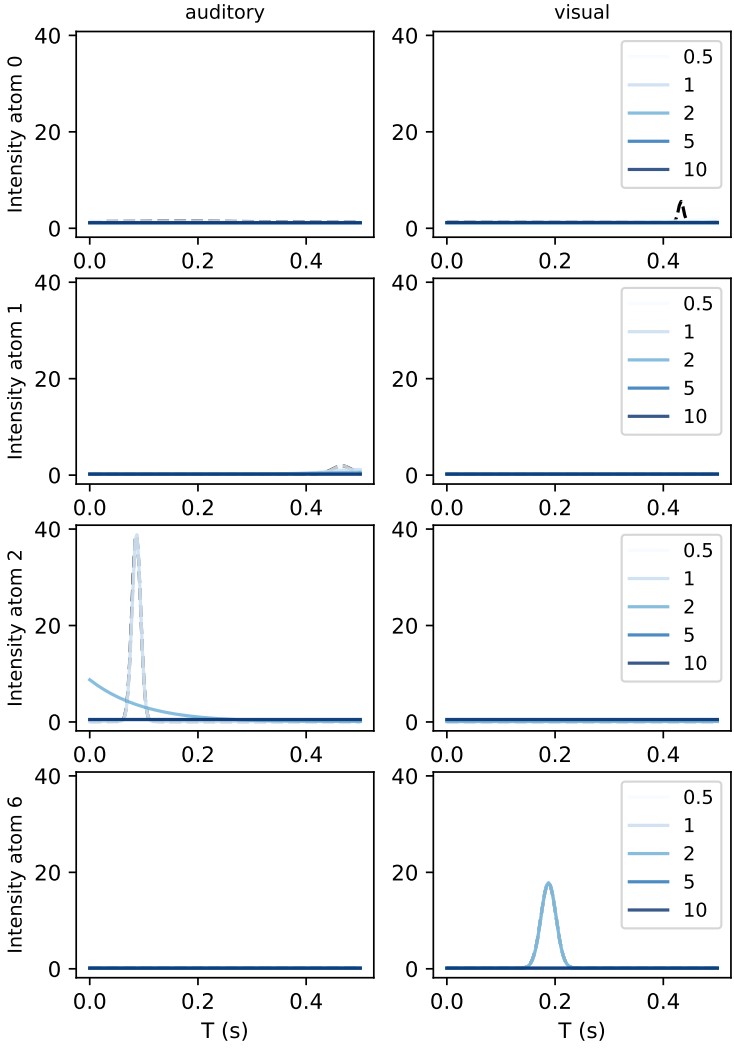

Figure A.4: Influence of the kernel truncation upper bound $b$ (with $a = 0$) on the obtained results on MNE sample dataset, for the 4 main atoms, for auditory and visual stimulus. By setting $b$ too high, all intensity functions are completely flat, indicating a total loss of information.

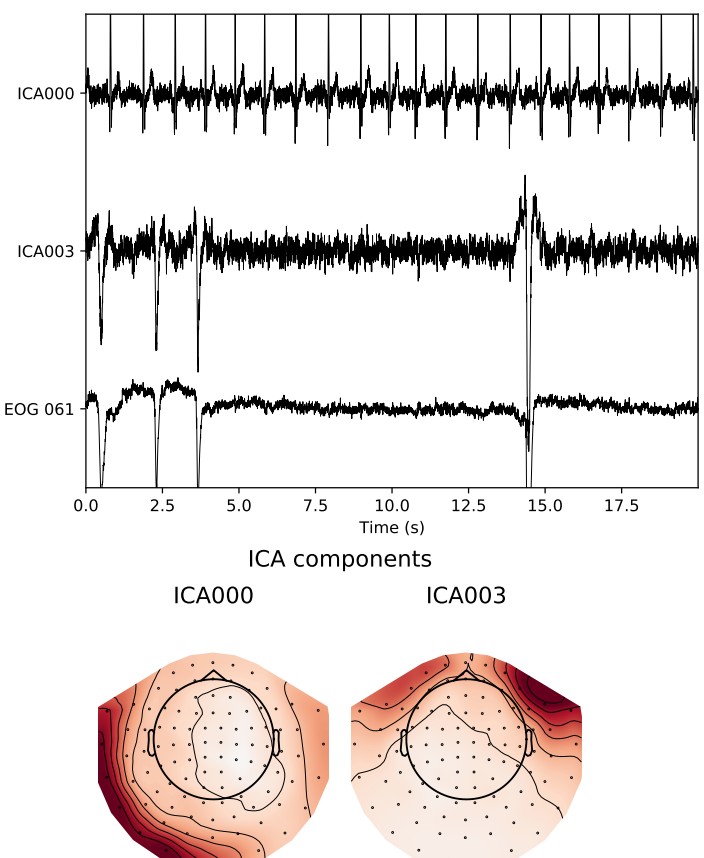

Figure A.5: The first two components alongside with electrooculography (EOG) signal (top) and their corresponding topomaps (bottom), on MNE *sample* dataset. `ICA000` can be associated to the heartbeat artifact, whereas `ICA003` can be associated to the eye-blink artifact.

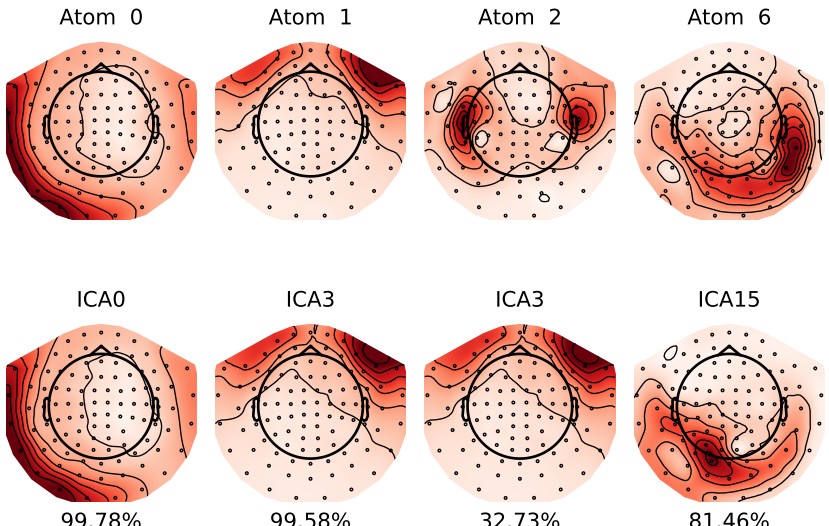

Figure A.6: Spatial representation of the four atoms presented in Figure 4 (top) alongside their ICA components with the maximum cosine similarity (value indicated at the bottom). Atoms and ICA are computed on MNE *sample* dataset.

evoked signals. Figure A.8 is similar but for the visual stimuli (again, stimuli on the left visual field and on the right visual field are combined).

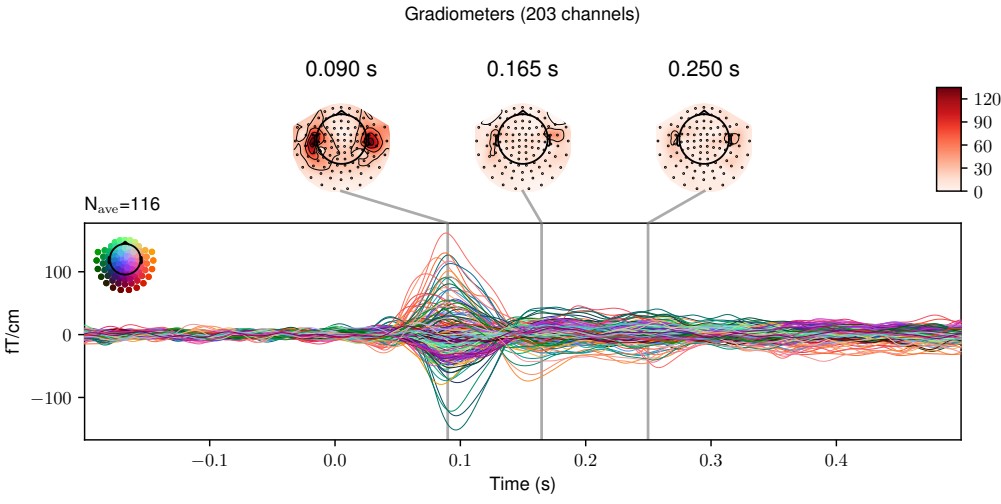

Figure A.7: Evoked signals following an auditory stimulus (cue at time = 0), on MNE *sample* dataset. Baseline correction applied from beginning of the data until time point zero.

On *somato* dataset, in order to exhibit the induced response to the somatosensory stimulus on the left median nerve of the subject, we perform a time/frequency analysis, presented on Figure A.9. In order to perform this analysis, a complex process including the use of Morlet wavelets is performed.

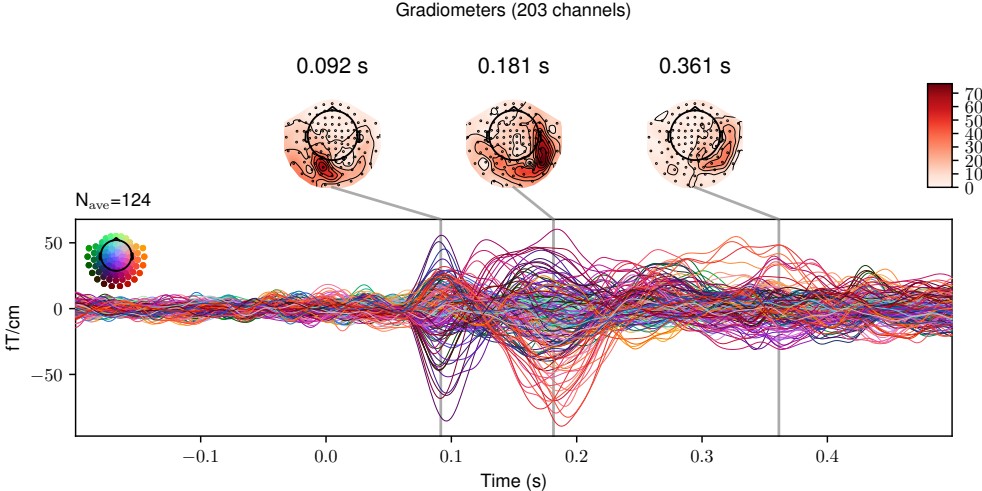

Figure A.8: Evoked signals following a visual stimulus (cue at time = 0) on MNE *sample* dataset. Baseline correction applied from beginning of the data until time point zero.

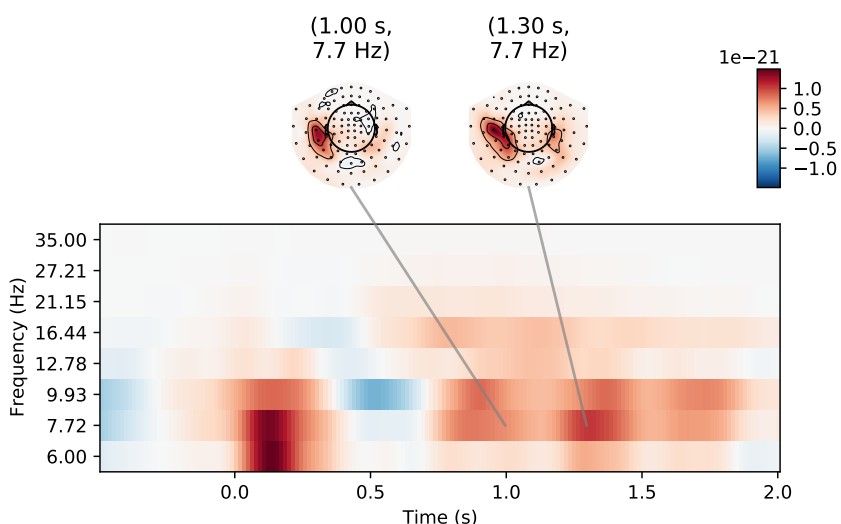

Figure A.9: Time-frequency plane for epoched signals following a somatosensory stimulus (cue at time = 0), on MNE *somato* dataset, with overviews at 1 and 1.3 seconds and 7.7 Hz. Baseline correction applied from time point -0.5 until time point zero.

Finally, note that these methods that are commonly used in the M/EEG data analysis comprise a thoughtfully data pre-processing as well as a manual intervention requiring domain knowledge to identify both artifacts and evoked and induced responses. As the proposed method in this paper is composed of a unified pipeline, it is a significant gain. Indeed, DriPP is able to automatically isolate artifacts without prior removal of artifacts using ICA, as well as capture the diversity of latencies corresponding to induced responses.

Regarding the statistical significance of the link between a stimulus and a neural pattern, one can consider performing a statistical test, where one wishes to reject the null hypothesis of independence.

| atom id | atom type | p-value auditory | p-value visual |
|---------|-----------|------------------|----------------|
| 0 | heartbeat | $2.25 \times 10^{-1}$ | $1.19 \times 10^{-1}$ |
| 1 | eye-blink | $5.85 \times 10^{-1}$ | $8.48 \times 10^{-1}$ |
| 2 | auditory | $2.31 \times 10^{-97}$ | $6.31 \times 10^{-1}$ |
| 6 | visual | $7.78 \times 10^{-1}$ | $5.12 \times 10^{-50}$ |

Table 1: On MNE sample dataset, statistical univariate Student t-test, $H_0$: independence between an atom and the stimulus (auditory or visual). The atom ids correspond to the ones presented in Figure 4.

We have performed a statistical test using a student t-test to check if the mean activation probability on $[a, b]$ segments is the same as the mean estimated on the baseline $[0, T] \setminus \bigcup_{t_i} [t_i + a, t_i + b]$. We present the results of the performed test in Table 1. We can clearly see that both artifacts (heartbeat and eye-blink) are not linked to stimuli while neural responses are linked to related stimuli (p-values are very small). We would like to stress that while this test is interesting, it is much more dependent on the selection of the support interval $[a, b]$ than the proposed method. Moreover, this test does not allow to assess the latency of the responses, and whether the atom is an induced response or an evoked one.

-

## A.6  EXPERIMENTS ON MNE SOMATO DATASET

We present in this section extra results obtained on the real dataset *somato*, that are complementary of Figure 5. Figure A.10 shows 3 atoms that correspond all to a $\mu$-wave located in the secondary somatosensory region (S2), with three different shapes of kernels in their estimated intensity functions.

## A.7  EXPERIMENTS ON CAM-CAN DATASET

The Cam-CAN dataset contains data of M/EEG recordings of 643 human subjects submitted to audio and visual stimuli. In this experiment, 120 bimodal audio/visual trials and eight unimodal trials – included to discourage strategic responding to one modality (four visual only and four auditory only) – are presented to each subject. For each bimodal trial, participants see two checkerboards presented to the left and right of a central fixation (34 milliseconds duration) and simultaneously hear a 300 milliseconds binaural tone at one of three frequencies (300, 600, or 1200 Hz, equal numbers of trials pseudorandomly ordered). For unimodal trials, participants either only hear a tone or see the checkerboards. For each trial, participants respond by pressing a button with their right index finger if they hear or see any stimuli (Shafto et al., 2014). For each subject, the experiment lasts less than 4 min.

The signals are pre-processed using low-pass filtering at 125 Hz to remove slow drifts in the data, and are resampled to 150 Hz to limit the atom size in the CDL. CDL is computed using `alphacsc` (Dupré la Tour et al., 2018) with the `GreedyCDL` method. Twenty atoms of duration 0.5 s are extracted from the signals. The extracted atoms' activations are binarized and, similarly as previous experiments, the events time are shifted to make them correspond to the peak amplitude time in the atom. Then, for every atom, the intensity function is estimated using the EM-based algorithm with 200 iterations and the "smart start" initialization strategy. Kernels' truncation values are set at $[0 \, \text{s} ; 0.9 \, \text{s}]$. Two drivers per atom are considered. The first driver contains timestamps of all bimodal trials (all frequencies combined) with in addition the 4 auditory unimodal stimuli (denoted `catch0`). The second driver is similar to the first one, but instead of the auditory unimodal stimuli, it contains the 4 visual unimodal stimuli (denoted `catch1`).

This experiment is performed for 3 subjects, for each one we plot the 5 atoms that have the highest ratio $\alpha/\mu$, so that we automatically exhibit atoms that are highly linked to the presented stimuli. Similarly as results presented in Section 4, we plot the spatial and temporal representation of each atom, as well as the two learned intensity functions that we plot "at kernel". Results are presented on Figure A.11, Figure A.12 and Figure A.13.

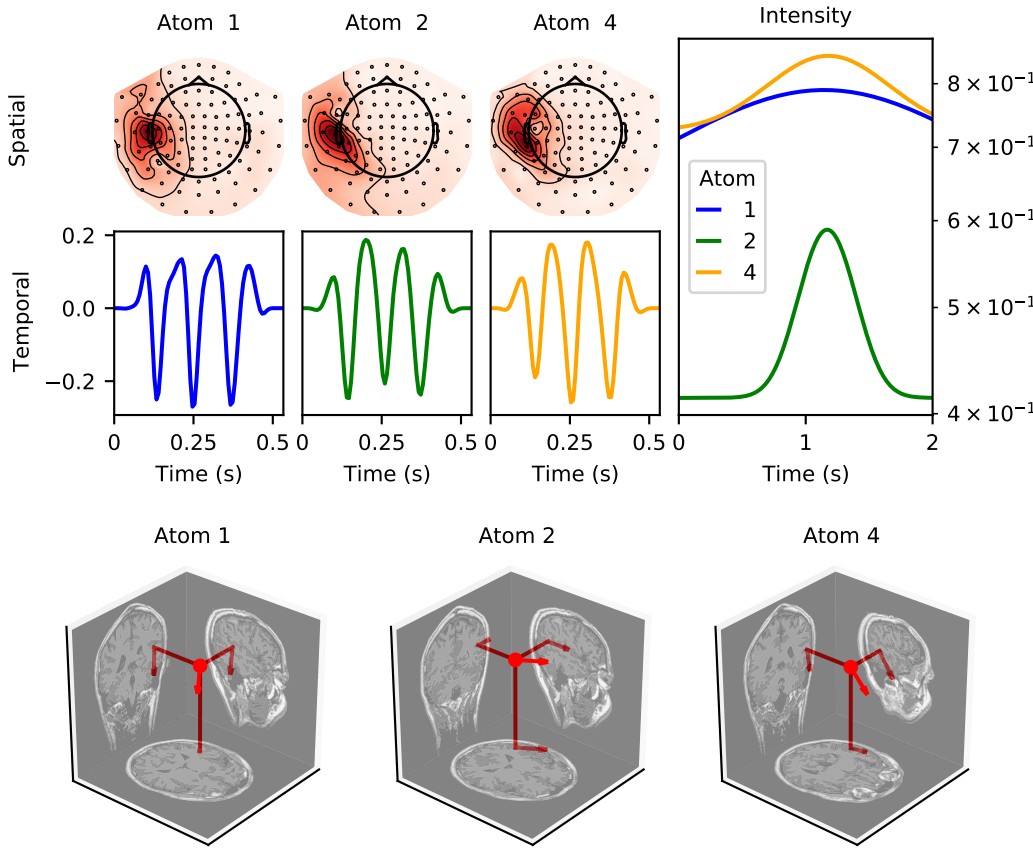

Figure A.10: Spatial and temporal patterns of 3 $\mu$-wave atoms from *somato* dataset, alongside with their respective estimated intensity functions. Below, in order to provide further information, are the corresponding brain locations for each atom obtained with dipole fitting.

First, we can observe that, as for experiments on *sample* and *somato* datasets, we recover the heartbeat artifact (atom 0 in Figure A.12) as well as the eye-blink artifact (atoms 0 in Figure A.11 and in Figure A.13). Because of the paradigm of the experiment, and in particular the instructions given to the subjects – that is, to press a button after every stimulus, the majority of them being visual –, it is not surprising if the eye blink artifact is slightly linked to the stimuli. Indeed, it is often observed in such experiments – where there is no designated time to blink – that the subject "allows" themself to blink after the visual stimulus.

As the majority of the presented stimuli are a combination of visual and auditory, the CDL model struggles to separate the two corresponding neural responses. Hence, that is why it can be observed that most of the first atoms reported exhibit a mixture between auditory and visual response in their topographies (first row). However, one can observe that for some such atoms, DriPP learned intensity is able to indicate what is the main contributing stimulus in the apparition of the atom. For instance, for atom 10 in Figure A.11, the auditory stimulus is the main responsible stimulus of the presence of this atom, despite this latter presenting a mixture of both neural responses. A similar analysis can be made for atom 1 in Figure A.12, but this time for the auditory stimulus.

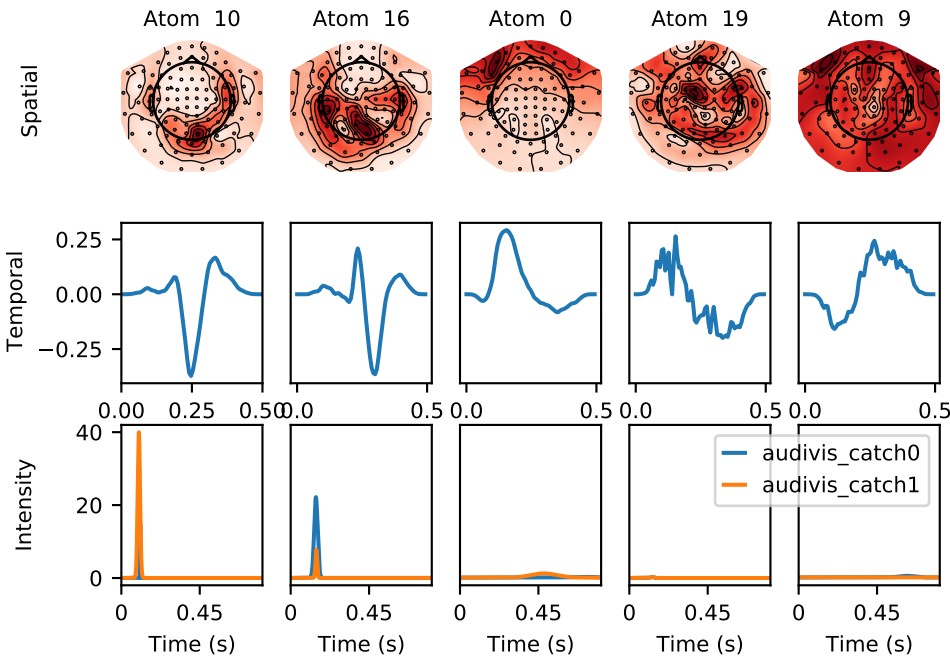

Figure A.11: Spatial and temporal patterns of the 5 atoms from Cam-CAN dataset subject CC620264, a 76.33-year-old female, and their respective estimated intensity functions following an audiovisual stimulus (cue at time = 0 s). Atoms are ordered by their bigger ratio $\alpha/\mu$.

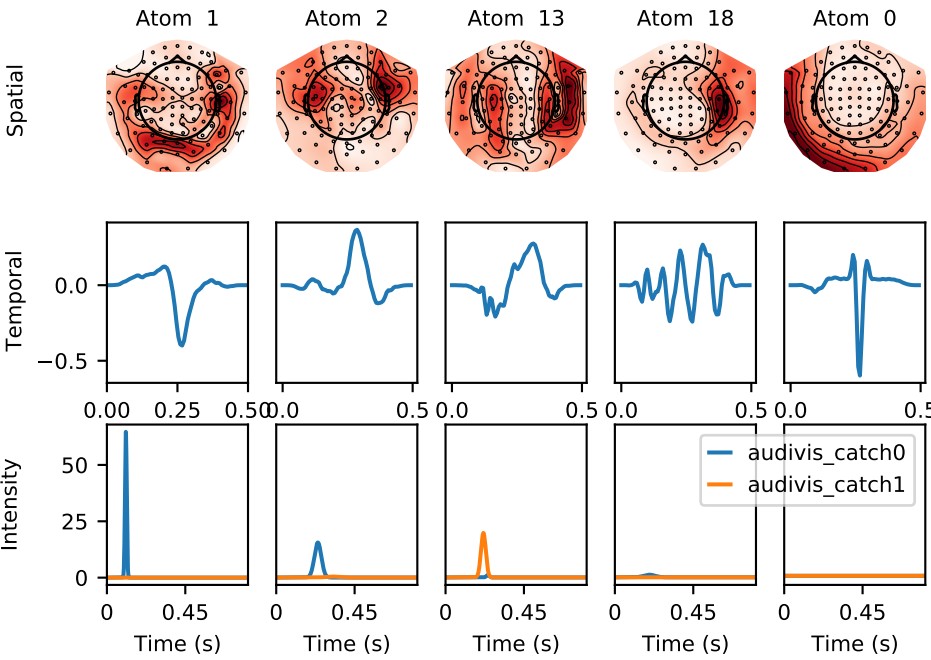

Figure A.12: Spatial and temporal patterns of the 5 atoms from Cam-CAN dataset subject CC520597, a 64.25-year-old male, and their respective estimated intensity functions following an audiovisual stimulus (cue at time = 0 s). Atoms are ordered by their bigger ratio $\alpha/\mu$.

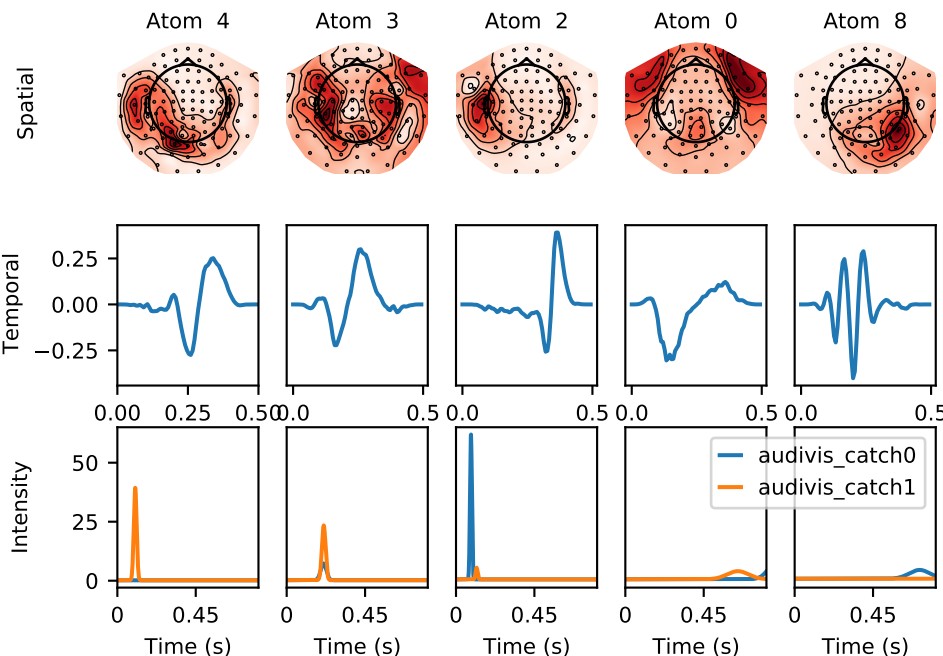

Figure A.13: Spatial and temporal patterns of the 5 atoms from Cam-CAN dataset subject CC723395, a 86.08-year-old female, and their respective estimated intensity functions following an audiovisual stimulus (cue at time = 0 s). Atoms are ordered by their bigger ratio $\alpha/\mu$.

