# OpenReview forum: "DriPP: Driven Point Processes to Model Stimuli Induced Patterns in M/EEG Signals"
_ICLR.cc/2022/Conference — ICLR 2022 Poster_

### Official Review · Reviewer_RDNE · 2021-10-30

**Correctness:** 3
**Technical Novelty And Significance:** 3
**Empirical Novelty And Significance:** 3
**Recommendation:** 6
**Confidence:** 4

**Main Review:**

1. The model is simple yet flexible. It captures the relative strength of the linkage through the ratio of the weight of the Gaussian and the baseline rate. The mean and standard deviation of the waveform occurrence timing relative to the stimuli timing is intuitive for neuroscientific experiments.

2. The approach seems to work well in practice. Most of the paper is clear and the figures illustrate the modeling approach.

3. The modeling approach avoids a 'tedious' approach of examining the rate of waveform occurrences in windows following the stimuli occurrence versus background cases (which could be done through point process hypothesis sampling with or without binning the waveform occurrences).

Concerns:
1. At times the manuscript could benefit from more theoretical or formal analysis as phenomenon are noted but not investigated: " This suggests that the estimators are slightly biased."

2. The pathological case at the beginning of step 5 seems speculative, rather than cast in a rigorous case for convergence.  Shouldn't $m \in [a,b]$ in order for the Gaussian model of jitter for the timing make sense? Furthermore, wouldn't it make sense to restrict $m\pm 3\sigma \in [a,b]$ for the same reason? If Gaussian jitter is not preferred why not consider other parametric families such as uniform or exponential distributions? These questions undermine the global choice of $a,b$ while $m$ and $\sigma$ are process and driver dependent. Later on page 7, it appears this choice is due to the inter-stimuli interval. Beside the mention of the degeneracy it is not clear why $m$ couldn't be restricted to lie in this range.

3.  The $\ell_\infty$ norm definition needs more clarification.  Equation 15 is a norm between functions on the real line, but its left-hand side definition implies the difference between parameter vectors.  It is not exactly a norm between intensity functions on the inside (which it could be using the known driver timings), but rather the maximum difference in intensity at various times since a driver event. This definition makes the y-axis of Figure 3 hard to interpret.

4. ICA overlap (cosine similarity) is mentioned for eye blink. Was this examined for the auditory or visually evoked responses?

5. Equation 7 the $\pi_{\mathbb{R}^+}$ function can be assumed to be a non-negative thresholding operating but it is not stated. Later after equation 10 it is stated as a projection operator to the ensemble $E$. Its not clear to me the choice of word ensemble, as compared to set.

6. I'd like to see a comparison of the proposed approach in determining whether or not a waveform is associated with a stimuli, versus using a test of independence (chi-squared) based on how many waveform occurrences fall into the window $[a,b]$ following a driver process.

7. The paper focuses on modeling the timing of waveforms already estimated through a different process from the original time-series. It would be useful to mention whether a joint model would make sense.  I can imagine that some original waveform occurrences could have been missed due to noise obscuring the waveform, but with the known driver process as well as the linking function additional occurrences may be able to be recovered.  Is this feasible?

Minor point:

"common techniques relies" -> "common techniques rely"




**Summary Of The Paper:**

The paper presents a model of the occurrence timing of waveforms in neural potential recordings in terms of a baseline rate and a set of driver processes. The waveforms and their timing are already extracted through a process of convolutional dictionary learning and convolutional sparse coding. The driver processes are known experimental stimuli timings. The driver to waveform model is done in a point process framework where the driver/stimuli increases the likelihood of the waveform in a window [a,b]. Specifically the paper proses a inhomogeneous Poisson process with a baseline rate and a truncated Gaussian distribution for each driver process.



**Summary Of The Review:**

The paper discusses a modeling approach relevant for understanding how waveforms in neural potential time series relate to known stimuli or background processes. The modeling approach seems consistent and performs well on the given data sets, but wider implications and insights as well as formal analysis are limited. Due to some minor concerns and large questions, my current opinion is borderline and I hope the authors can help clarify any issues raised.

++++++++++++++++++++++=
Update after revision:
I think my minor concerns and questions were answered. I encourage the authors to carefully proofread the revision.

My concern with not raising the score more are the limited scope/impact of the work in the learning representation community. Nonetheless, I think that meaningful practical utility may be gained in neuroscience analysis from the proposed approach.

---

### Official Review · Reviewer_ZvAC · 2021-11-02

**Correctness:** 4
**Technical Novelty And Significance:** 4
**Empirical Novelty And Significance:** 4
**Recommendation:** 8
**Confidence:** 4

**Main Review:**

The methodology is introduced precisely. The inspiring application of the proposed DriPP technique wonderfully extracts peripheral (ECG) and brain-related (ERP components), which are crucial for the proper analysis of M/EEG. The illustrative examples are apparent, and provided code works fine. The authors stated that the current approach utilizes some domain knowledge to set hyperparameter values in their limitations. An exploration of data-driven strategies to infer these hyperparameters shall follow. It would also be interesting to show whether, for example, the extracted ECG/heartbeat atom is periodic after extending the analysis window. A similarity of the proposed method to old-school ICA is somehow disappointing. The authors shall explain this part in more detail since the limitations of ICA applications are somewhat known in the community.

**Summary Of The Paper:**

This is an exciting manuscript. It was a delight to read and evaluate it. The manuscript is well written, and the methodology is presented and validated with illustrative cases. Provided code runs without problems, and it is excellently documented. An introduced point process-based technique seems to be adequately fitted to model brain external/reactive and induced/active/imagery experiments. An influence on M/EEG captured spatiotemporal patterns appears to be also straightforward. The introduced technique extracts spatiotemporal patterns through the utilization of convolutional dictionary learning. The proposed parameterization procedures work thoroughly. The authors clearly explain the limitations of the current approach as related to the present preliminary application.

**Summary Of The Review:**

To summarize, the authors propose an interesting DriPP technique to decompose complex M/EEG signals with a bit disappointing twist. The resulting components are 95% similar to what ICA could do, and the ICA is scrutinized at the beginning of the submission, which seems confusing and disappointing.

---

### Official Review · Reviewer_87RT · 2021-11-02

**Correctness:** 2
**Technical Novelty And Significance:** 3
**Empirical Novelty And Significance:** 3
**Recommendation:** 3
**Confidence:** 3

**Main Review:**

The paper introduces an interesting technique for the characterization of neuronal events that can be acquired on E/MEG signals. Such an approach could help better understanding and characterization these signals, which contain important information but whose analysis highly depend on the clinician and the analysis technique used.
I have however several main issues with the study
The authors do not seem to compare their technique with any approach from the state-of-the-art, and it is therefore quite hard to judge the performance of the technique.
The technique requires the use of several pre-processing technique, and most importantly an event detection technique (the authors suggested and used a CDL for this task). Have the authors assessed how the DriPP performance is dependent on the quality of the event detection?
The authors do provide a single quantitative assessment of the technique, but results assembled on figure 3 does not allow the reader to understand which sets of parameters to choose, or what would be an acceptable error.
The authors suggest a smart initialization procedure, which is used for all experiments but the effect or benefit of this procedure has not been evaluated.
The notation used for the description of the DriPP is quite confusing, and in equation 2, I am not sure to distinguish the sets of timepoints (t_i^k) from A_k and T_p.


**Summary Of The Paper:**

This paper introduces a novel technique for the analysis of electro or magneto encephalogram based on an innovative point process approach. The point process model is estimated using an expectation maximization algorithm and is aimed at characterising events in the signals representing the brain activity. The technique is finally evaluated on synthetic data and real data from openly available databases.

**Summary Of The Review:**

The paper introduces an interesting technique for the characterization of neuronal events, but I do feel that the technique should be evaluated more thoroughly and compared with other approaches.

---

### Official Review · Reviewer_vbHx · 2021-11-05

**Correctness:** 3
**Technical Novelty And Significance:** 3
**Empirical Novelty And Significance:** 3
**Recommendation:** 8
**Confidence:** 4

**Main Review:**

Strengths
-	Inclusion of results from synthetic and empirical data to demonstrate the effectiveness of their method. Inclusion of results from evoked and induced responses support the generalizability of the DriPP model.
-	Overall, the paper is very well written and illustrated. Authors provide some details to reproduce experiments.

Weakness
-  A comparison with other related work (e.g., epoch averaging) is needed to support authors claims that their work addresses limitations of these prior work. There appears to be some similarities with ICA, which is given limited consideration in the text.
-   Results presented are from limited number of subjects and no quantitative measures are provided to assess model performance. It should be noted that authors include results from subjects from another dataset in the supplementary material, but reviewer relied more on results presented in the main text to make assessment.
-	It is not clear why some model choices are made (e.g., Gaussian kernel, number of atoms, "smart start" initialization) and how domain knowledge is used to determine some model parameters (e.g., kernel hyperparameter, authors acknowledge this limitation). Selection of kernel supports seems arbitrary (authors suggest based on ISI).
-	How is the number of atoms and the duration of atoms determined for each dataset? (20 for somato and 40 for sample)
       “For the sample dataset, 40 atoms of duration 1 s each are extracted, and for the somato dataset 20 atoms of duration 0:53 s are estimated.”
-	What does the latency of atom 6 (Figure 4) correspond to?
“Regarding atom 6, topography is right lateralized in the occipital region suggesting a visual evoked reponse. This is confirmed by the intensity function estimated that reports a relationship between this atoms and the visual stimuli.”
-	Model considers only excitatory effects and not inhibitory effects (authors acknowledge this limitation).
-	To model a PP, only stimulus onset is considered. So potential for DriPP model to be limited in scenarios with continuous stimuli.

Additional Feedback
-	Have a transition at the end of the introduction to state the main contributions of the paper and the overall goal of the experiment.
-	Figure 1: Define CDL in caption
-	Figure 3: More informative caption. Label the legend.
-	Figure 5: Specify that atom 0 is related to eye blink. What of atom 7? Typo: An somatosensory stimulus will induced
-	Describe the N100 response to relate to the stimulus.
        “This is the N100 response well known in the MEG/EEG literature.”
-	Authors provide justification for the choice of the triggering kernel (Gaussian) in the Discussion. This should be moved to earlier in the text when the kernel is first mentioned.
-	Fix typos (not an exhaustive list):
        “Regarding atom 6, topography is right lateralized in the occipital region suggesting a visual evoked reponse. This is confirmed by the intensity function estimated that reports a relationship between this atoms and the visual stimuli.”



**Summary Of The Paper:**

This paper presents a novel model for revealing underlying stimulus-response patterns in MEG/EEG data. The main contributions include a driven temporal point process (DriPP) model that leverages prior work on convolutional dictionary learning (CDL) and point processes to be amenable to MEG/EEG data, an EM algorithm for inferring the DriPP model parameters and application of the model to synthetic data and empirical data.

**Summary Of The Review:**

The proposed DriPP model is broadly applicable to neuroscience applications to isolate underlying stimulus-driven neural patterns and unrelated temporal patterns (background neural activity, artifacts, etc.).

Authors need to provide justifications to support model parameter choices. A comparison with other related work (e.g., epoch averaging) is needed to support authors claims that their work address limitations of these prior work. There appears to be some similarities with ICA, which is given limited consideration in the text.

-------------------
Update post-rebuttal: Reviewer read the author responses and the revised manuscript, as well as other reviewer comments and respective responses. Appreciated the author responses and revisions to the issues raised by this reviewer and other reviewers, and most of the main critiques are addressed. Potential to improve/extend DriPP method in future work. There is an applications track (neuroscience) in this conference, and this work is of relevance to the broader learning representations community. Revising score upwards. Authors should proof-read the manuscript for typos (also see other reviewer comments).

---

### Decision · Program_Chairs · 2022-01-20

**Decision:**

Accept (Poster)

**Comment:**

This paper presents a novel method for identifying simuli induced
patterns in MEG and EEG signals.  The authors develop a novel statistical
point process model and a fast EM algorithm to learn the parameters.

Discussion of this paper centered around: how to fit hyperparameters,
and similarity and comparison with other algorithms, especially ICA, as
well as the small number of subjects

Comparison to other methods would make the work stronger, as would
adding more datasets but this novel algorithm seems worth publishing.
I recommend acceptance as a poster.